# Expression and Variations in *EPAS1* Associated with Oxygen Metabolism in Sheep

**DOI:** 10.3390/genes13101871

**Published:** 2022-10-15

**Authors:** Qiming Xi, Fangfang Zhao, Jiang Hu, Jiqing Wang, Xiu Liu, Pengju Dang, Yuzhu Luo, Shaobin Li

**Affiliations:** 1Gansu Key Laboratory of Herbivorous Animal Biotechnology, Faculty of Animal Science and Technology, Gansu Agricultural University, Lanzhou 730070, China; 2International Science and Technology Cooperation Base of Meat Sheep and Meat Cattle Genetic Improvement in Northwest of China, Gansu Agricultural University, Lanzhou 730070, China; 3Gansu Agriculture Technology College, Lanzhou 730070, China

**Keywords:** endothelial PAS domain protein 1, sheep, oxygen metabolism, partial pressure of oxygen, blood oxygen saturation

## Abstract

Endothelial PAS domain protein 1 gene (*EPAS1*) is a member of the HIF gene family. This gene encodes a transcription factor subunit that is involved in the induction of oxygen-regulated genes. Several studies have demonstrated that a mutation in *EPAS1* could affect oxygen sensing, polycythemia, and hemoglobin level. However, whether *EPAS1* mutation affects sheep oxygen metabolism is still unknown. Therefore, we explored the relationship between the variation of *EPAS1* and oxygen metabolism in sheep. In this study, variations in ovine *EPAS1* exon 15 were investigated in 332 Tibetan sheep and 339 Hu sheep by polymerase chain reaction-single strand conformation polymorphism (PCR-SSCP) analysis. In addition, we studied the effect of these variations on blood gas in 176 Tibetan sheep and 231 Hu sheep. Finally, the mRNA expression of *EPAS1* in six tissues of Hu sheep and Tibetan sheep living at different altitudes (2500 m, 3500 m, and 4500 m) was analyzed by real-time quantitative PCR (RT-qPCR). Four alleles (*A*, *B*, *C*, and *D*) were detected, and their distributions highly differed between Tibetan sheep and Hu sheep. In Tibetan sheep, *B* was the dominant allele, and *C* and *D* alleles were rare, whereas all four alleles were common in Hu sheep. Six single nucleotide polymorphisms (SNPs) were identified between the four alleles and one of them was non-synonymous (p.F606L). While studying the blood gas levels in Tibetan sheep and Hu sheep, one variant region was found to be associated with an elevated *p*O_2_ and sO_2_, which suggested that variations in *EPAS1* are associated with oxygen metabolism in sheep. RT-qPCR results showed that *EPAS1* was expressed in the six tissues of Hu sheep and Tibetan sheep at different altitudes. In addition, the expression of *EPAS1* in four tissues (heart, liver, spleen, and longissimus dorsi muscle) of Hu sheep was lower than that in Tibetan sheep from three different altitudes, and the expression of *EPAS1* was positively correlated with the altitude. These results indicate that the variations and expression of *EPAS1* is closely related to oxygen metabolism.

## 1. Introduction

Hypoxia-inducible factors (HIFs) are response regulators that mediate the adaptive response of cells under hypoxic or low oxygen conditions and regulate gene expression. HIFs are composed of two subunits, namely, α and β, of which α is the functional subunit (including 1α, 2α, and 3α) and β is the structural subunit [1,2]. Endothelial PAS domain protein 1 gene (*EPAS1*) is a member of the HIF gene family, which encodes hypoxia-inducible factor 2α (HIF-2α). *EPAS1* induces the expression of oxygen-regulating genes under hypoxic conditions (when oxygen concentrations decrease) [3]. HIF-2α contains a basic helix-loop-helix (hHLH) and Per-ARNT-Sim (PAS) domains that respond to oxygen levels [4]. *EPAS1* is involved in the development of the embryonic heart and contributes to the adaptation of highland indigenous populations and animals to the hypoxic environment of the plateau [5,6]. As a native species of the Tibetan Plateau, Tibetan sheep well adapt to plateau hypoxia. Compared with the blood gas indexes in lowland Hu sheep, the blood gas indexes of Tibetan sheep were found to be different, and the HIF signaling pathway was activated [7].

Several studies have reported that genetic variation in *EPAS1* is related to human polycythemia and high-altitude hypoxia adaptation. One of the studies reported that SNPs in *EPAS1* affected human erythrocyte count and hemoglobin volume; however, no effect on oxygen saturation [8] was observed. Variations in the *EPAS1* gene allow Sherpas to adapt to high-altitude hypoxia [9]. *EPAS1* variants are associated with multiple physiological traits of Tibetans [10]. However, whether the mutation in *EPAS1* affects the oxygen metabolism in sheep is still unknown.

A study on Tibetan sheep’s adaptation to high altitude revealed that variation in exon15 of *EPAS1* was rich [7]. We investigated the potential variants in exon 15 of *EPAS1* of the plateau breed Tibetan sheep and the plain breed Hu sheep using polymerase chain reaction-single strand conformation polymorphism (PCR-SSCP) analysis. Subsequently, the effect of the genetic variation on blood gas was investigated. Finally, the expression level of *EPAS1* in sheep was analyzed.

## 2. Materials and Methods

### 2.1. Experimental Animals and Data Collection

Variations in the *EPAS1* gene were analyzed in 339 Hu sheep and 332 Tibetan sheep using the PCR-SSCP technique. Among these sheep, 176 Tibetan sheep (134 female, 42 male) and 231 Hu sheep (200 females, 31 males) were used for blood gas analysis. All these sheep were about 3.5 years old and healthy. The Tibetan sheep live at an altitude of over 2800 m in Maqu County, Gansu Province, China, and were obtained from a herdsman’s flock (about 1000). The Hu sheep was from the Zhejiang Hongtai Agriculture and Animal Husbandry Technology Co., Ltd., at an altitude of less 100 m, in Yuqian Town, Zhejiang Province, China. Blood was collected from the jugular vein of each sheep using a 5-mL sodium heparin collection tube. A small sample of the blood was used for blood gas analysis and the remaining was collected on the Munktell TFN paper (Munktell Filter AB, Falun, Sweden) for DNA purification [11].

For *EPAS1* expression analysis, nine Tibetan sheep living at different altitudes (different oxygen concentrations) and three Hu sheep were randomly selected. These Tibetan sheep were selected from Zhuoni County, Gansu Province, China (2500 m), Haiyan County, Qinghai Province, China (3500 m), and Zhiduo County, Qinghai Province, China (4500 m), with three sheep at each altitude selected randomly from the herders’ flock. The three Hu sheep were from Kangrui Breeding Sheep Co., Ltd. in Baiyin City, Gansu Province, China (1800 m). All sheep were 3.5 year-old ewes with good health. Since the male sheep of the same age in the group were breeding rams, only ewes could be selected for slaughter. Each experimental sheep was euthanized by intravenous injection of pentobarbital sodium (350 mg), with cardiac arrest and continuous involuntary breathing for about 3 min without blinking reflex. The sheep were then dissected, and the heart, liver, spleen, lungs, kidneys, and longissimus dorsi were collected for RT-qPCR. The collected tissue samples were immediately placed in liquid nitrogen and transferred to the laboratory, and then stored at −80 °C to extract the total RNA.

### 2.2. Blood Gas Indicator Measurement

Blood gas biochemical indexes were measured directly using the i-STAT blood gas analyzer (Abbott, Chicago, IL, USA), including partial pressure of oxygen (*p*O_2_), partial pressure of carbon dioxide (*p*CO_2_), pH, and blood oxygen saturation (sO_2_). In addition, the partial pressure of oxygen at which hemoglobin is 50% saturated with oxygen (*p*50) was calculated using *p*O_2_, sO_2_, and pH [12,13]. The following equations were used.
$$p_{50}\;std=anti\;log\frac{log\left(\frac{1}{k}\right)}{n}\;;\mathrm{where}\;\frac{1}{k}=\left[anti\log(n\log pO_{2\left(7.4\right)})\right]\cdot \frac{100-sO_{2}}{sO_{2}}$$

A Hill constant “*n*” for hemoglobin A of 2.7 was used. The *p*O_2_ in venous blood at 37 °C was converted to *p*O_2_ at pH 7.4 using the following equation:$$\log pO_{2\left(7.4\right)}=\log pO_{2}-\left[0.5\left(7.40-\mathrm{pH}\right)\right]$$

### 2.3. Primers for PCR and RT-qPCR

One pair of PCR primers was designed to amplify a 470 bp fragment containing exon 15 and part of intron 14 of ovine *EPAS1*. A pair of primers was designed for *EPAS1* expression, referring to the sequence of sheep *EPAS1* gene (XM_015094403) published by GenBank. The primers were designed using the Premier 5.0 software (Table 1). β-actin (accession number: NM_001009784) was used as the reference gene. The primers were synthesized by Wuhan Aokedingsheng Biotechnology Co., Ltd. (Wuhan, China).

### 2.4. Amplification of Ovine EPAS1

PCR amplifications were performed using S1000 thermal cyclers (BioRad, Hercules, CA, USA), and were performed in a 20-µL reaction containing the purified genomic DNA on a 1.2-mm punch of the FTA paper, 0.25 μM of each primer, 0.5 U of Taq DNA polymerase (Takara, Dalian, China), 150 μM of per dNTP (Takara), 2.5 mM Mg^2+^, 2.0 μL of 10× PCR buffer (supplied with the DNA polymerase enzyme) and deionized water to make up the volume to 20 μL. The amplification steps were 2 min at 94 °C, followed by 35 cycles of 30 s at 94 °C, 30 s at 61 °C, and 30 s at 72 °C with a final extension of 5 min at 72 °C.

### 2.5. Screening for Variation and Sequencing of Variants

The PCR amplicons were screened for variations in sequences using the SSCP method. For each amplicon, a 0.7-μL aliquot was mixed with 7 μL of the loading dye (0.025% bromophenol blue, 98% formamide, 0.025% xylene cyanol, 10 mM EDTA). Samples were denatured at 95 °C for 5 min, cooled rapidly on wet ice, and subsequently loaded onto 16 cm × 18 cm, 14% acrylamide:bisacrylamide (37.5:1) (Bio-Rad) gels. Electrophoresis was performed for 20 h in 0.5 TBE at 220 V and 27 °C, and gels were silver stained [14].

PCR-SSCP analysis revealed these amplicons to be homozygous, and these were sequenced in both directions using the Sanger sequencing approach at Sangon Biotech Company Limited, Shanghai, China. For heterozygous amplicons, we used an alternative method for sequencing [15]. Sequence alignments, translations, and comparisons were performed using DNAMAN (version 8.0.8, Lynnon BioSoft, Vaudreuil, QC, Canada).

### 2.6. Real-Time Quantitative PCR (RT-qPCR) Analysis

The total RNA was isolated from the six tissues of Hu sheep and Tibetan sheep using the Trizol reagent (Shanghai Yuanye Biotechnology Co., Ltd., Shanghai, China) according to the manufacturer’s instructions. The purity and concentration of total RNA were monitored using an ultraviolet spectrophotometer. RNA reverse transcription was performed using the instructions provided in the kit (Nanjing Novizan Biotechnology Co., Ltd., Nanjing, China).

RT-qPCR was performed using the SYBR Green Pro Taq HS qPCR Kit (Accurate Biology, Human, China) with cDNA as the template. RT-qPCR amplification was performed in a reaction mixture of 20 μL, containing 7.2 μL RNase-free water, 10 μL 2× Premix (AG, Changsha, China), 0.4 μL each for the forward and reverse primers (10 ng/μL), and 2 μL of cDNA (100 ng/μL). The thermal cycle parameters are as follows: initial denaturation at 95 °C for 30 s, followed by 40 cycles of 95 °C for 5 s and 60 °C for 30 s. The melting temperature (Tm) peak at 85 °C ± 0.8 on the dissociation curve was used to determine the specificity of PCR amplification. The 2^−Δ^^Δ^^CT^ method [16] was used for analyzing the relative expression of *EPAS1*. Calculation method: first, for all test samples and calibration samples, the C_T_ values of reference genes were normalized to the C_T_ values of target genes: ΔC_T_ (test) = C_T_ (target, test) − C_T_ (ref, test); ΔC_T_ (calibrator) = C_T_ (target, calibrator) − C_T_ (ref, calibrator). Second, the ΔC_T_ value of the calibration sample was normalized to the ΔC_T_ value of the test sample: ΔΔC_T_ = ΔC_T_ (test) − ΔC_T_ (calibrator). Finally, the expression ratio was calculated as follows: 2^−ΔΔCT^ = ratio of expression quantity. After the qPCR reaction, the standard curve was built by taking the log value of dilution multiple of template series as the *X*-axis and the corresponding C_T_ value as the *Y*-axis (or vice versa). We used three biological replicates and three technical replicates to ensure that the findings were genuine and trustworthy, and all the STD of Ct between technical replicates were less than 0.2.

### 2.7. Statistical Analyses

Sheep with incomplete data were removed from certain analyses so that the sample size varied in these analyses. Statistical analyses were performed using Minitab version 17 (Minitab, Inc., State College, PA, USA). General linear mixed-effects models (GLMMs) were used to evaluate the effect of the absence or presence (coded as 0 or 1, respectively) of *EPAS1* variants on the blood gas levels. Breed and gender were used as fixed factors in the model to evaluate the effect of gene variation on *p*CO_2_, and the breed was used as the fixed factor in the model for other association analyses. Only primary effects were tested in this experiment, and the correlation was considered significant at the 5% level. The single factor analysis of variance (ANOVA) in SPSS 22.0 statistical software was used to analyze the RT-qPCR results. The data are expressed as mean ± standard errors of the mean, and *p* < 0.05 indicated significant differences.

## 3. Results

### 3.1. Variation in Ovine EPAS1

We detected four unique PCR-SSCP banding patterns in ovine *EPAS1* exon 15, with either one or a combination of two banding patterns observed in each sheep (Figure 1). DNA sequencing results showed that these PCR-SSCP patterns represented four variant sequences of *EPAS1* (*A*, *B*, *C*, and *D*). Six single-nucleotide polymorphisms (SNPs) were identified, five of which (c.1641G > A, c.1674G > C, c.1816T > C, c.1824C > T, and c.2028G > A) were present in the exon region, whereas only one of them (c.1816T > C) was a non-synonymous mutation, resulting in an amino acid change (p.F606L) (Table 2).

### 3.2. Comparison of Allele and Genotype Frequencies between Tibetan Sheep and Hu Sheep

The following 10 genotypes were detected in Hu sheep: *AA*, *AB*, *AC*, *AD*, *BB*, *BC*, *BD*, *CC*, *CD*, and *DD*, whereas only 8 of them were found in Tibetan sheep: *AA* (0.30%), *AB* (7.23%), *AC* (0.90%), *AD* (0.30%), *BB* (83.13%), *BC* (3.31%), *BD* (4.22%), and *CD* (0.60%). In Tibetan sheep, *BB* was the dominant genotype, *B* was the dominant allele, *C* and *D* were rare, whereas the four alleles were common in Hu sheep (Table 3).

### 3.3. Effect of Breed and Gender on Blood Gas in Sheep

The breed exerted a significant effect on three blood gas (*p*CO_2_, *p*O_2_, and sO_2_) (*p* < 0.01). In addition, gender exerted a significant effect on *p*CO_2_ (*p* < 0.01, Table 4). Afterward, we evaluated the effect of the genetic variation on *p*CO_2_, the breed and gender were used as fixed factors in the model, and the other association analysis breed was fitted as the fixed factor in the model.

### 3.4. Effect of Variation in EPAS1 on Blood Gas

The presence of variant *A* was found to be associated with increased values of *p*O_2_ and sO_2_. No association was found between *B, C,* and *D* variants and four blood gas (*p*CO_2_, *p*O_2_, sO_2_, and *p*_50_) in sheep (Table 5).

### 3.5. Expression of EPAS1 in Hu Sheep and Tibetan Sheep from Different Altitudes

The RT-qPCR results revealed that *EPAS1* was expressed in six tissues of Hu sheep and Tibetan sheep from different altitudes. The expression in the heart, liver, spleen, and longissimus dorsi muscle increased with increasing altitude, and the expression in these four tissues of all Tibetan sheep was significantly higher than that of Hu sheep. No significant difference was observed in the expression in the lung and kidney tissues between 2500 m Tibetan sheep and Hu sheep; however, a significantly increased was observed in the expression of six tissues in 4500 m Tibetan sheep. Among the six tissues of Tibetan sheep at three different altitude gradients, the expression of *EPAS1* was the highest in the heart (Figure 2). These results indicated that the expression of *EPAS1* was related to the oxygen concentration; a lower oxygen concentration resulted in its higher levels.

## 4. Discussion

Oxygen is an essential substrate for life activities, but its concentration gradually decreased with the altitude increase. To cope with the low oxygen concentration at high altitude, phenotypic and genetic alterations have occurred in Tibetan sheep. One of their hypoxia response strategies is to change the hematocrit and hemoglobin concentration in the blood [17], which will induce a change of blood gas that is mainly determined by the adapted number of red cells. And its inheritance is worth further exploration.

In this study, the relationship between expression and variation of *EPAS1* and blood gas in sheep was studied. HIFs are activated and function as transcriptional regulators of genes involved in the hypoxic response [3,18]. Variations in *EPAS1* may affect the gene expression and relate to functional changes, which have been observed in the adaptation of humans and animals to high altitudes [18,19,20], certain human disorders [21,22], and embryonic development. To our knowledge, this is the first study on the relationship between *EPAS1* variants and blood gas in sheep.

Five SNPs were detected in exon 15 of ovine *EPAS1*, compared to a previous study that detected four SNPs in exon 15 of Tibetan sheep (*n* = 30) and Mongolian sheep (*n* = 31) using Sanger sequencing [7]. Among these, three (c.1641G > A, c.1816T > C, c.1824C > T) were the same as that observed in the current study. We speculate that more variations will be found if more sheep of more breeds are investigated. Tibetan sheep are a typical indigenous animal in the Qinghai-Tibet plateau with extreme living conditions (high altitude, low temperatures, and low oxygen). The oxygen concentration gradually decreases with increasing altitude, with oxygen levels at 4000 m above sea level being only 60% of that at sea level [23]. Lower oxygen partial pressure can lead to insufficient oxygen supply to body tissues, thereby affecting the normal physiological functions of animals [24]. Studies have demonstrated that extreme living conditions (hypoxic stress at high altitudes) can alter the diversity of functional genes relative to fitness traits through natural selection, resulting in an increased frequency of favorable mutations. The frequency of these mutations increases in subsequent generations [25,26]. In this study, the distribution of four alleles highly varied between Tibetan sheep and Hu sheep, and similar results were obtained in a study on the distribution of *EPAS1* intermediate genes in Tibetan and Han populations [13,27], explaining the large difference in allele distribution between the two sheep breeds.

Over 97% of Tibetan sheep were *B* carriers, suggesting that variant *B* may have a good response to bad conditions on the plateau. Although variant *B* did not affect blood gas investigated in this study, it may affect oxygen utilization in another way. Functional assays showed that certain variants led to higher stability and heterodimerization activity of the *EPAS1*, as well as low volume of red blood cells, high hemoglobin levels, and increased anaerobic metabolism capacities [20]. Thus, the effect of variant *B* on the above indicators needs to be further studied.

Zhao et al. reported significant differences in blood gas indices between low-altitude breeds (Hu sheep) and high-altitude breeds (Tibetan sheep and Gansu Alpine Merino sheep) and concluded that the differences in blood gas indices were primarily caused by altitude differences [17]. In this study, variant *A* was associated with *p*O_2_ and sO_2_. Moreover, *p*O_2_ was proportional to dissolved oxygen concentration and directly measures the oxygen available to cells. The sO_2_ quantifies the amount of oxygen carried by blood hemoglobin. Both *p*O_2_ and sO_2_ are important hemodynamic parameters of oxygen metabolism [28], suggesting that the mutation in *EPAS1* affected oxygen metabolism in sheep.

The effect of variant *A* could be associated with the SNP (c.1824), which was located near the topologically associating domains (TAD) of HIF-2α; TAD plays a leading role in the specific regulation of different target genes [29]. Although the SNP (c.1824) was synonymous and would not lead to amino acid substitutions, it may affect the expression or structure of the protein. Similarly, silencing mutations could affect the rate of mRNA translation, thereby altering the folding of proteins [30]. We believe the effects observed in the study could also be attributed to the SNP linked to sequence variations in other regions of the gene that regulates gene expression or function.

*EPAS1* plays an important role in the adaptation of animals to hypoxia [31]. VHL (a combination of CNS hemangioblastoma and renal or pancreatic cysts, pheochromocytoma, renal cancer, and ectodermal cystadenoma and other diseases) inactivation in the zebrafish stabilizes *HIF-1α* and *EPAS1* with consequent upregulation of specific target genes involved in cell proliferation, angiogenesis, and erythropoiesis, increasing the ability of the blood to transport oxygen [32]. At present, the research on the expression of *EPAS1* is primarily concentrated in the medical field, especially in tumors, whereas the research on Tibetan sheep adaptation to the plateau environment is sparse [33]. In this study, the expression of *EPAS1* was detected for the first time in six tissues of Tibetan sheep at three elevation gradients (2500 m, 3500 m, and 4500 m), and was compared with the expression of related tissues of Hu sheep. The results showed that *EPAS1* was expressed in all six tissues, indicating its wide expression in the tissue. In addition, the expression of *EPAS1* in four tissues (heart, liver, spleen, and longissimus dorsi muscle) of Hu sheep was lower than that in Tibetan sheep from three different altitudes, and the expression of *EPAS1* was positively correlated with the altitude. Our previous study showed that the *p*O_2_, *p*CO_2_, and sO_2_ in blood of Tibetan sheep from three altitudes (2500 m, 3500 m, and 4500 m) would be significantly reduced with the altitude increase [24]. This suggests that Tibetan sheep would increase the expression of *EPAS1* under hypoxia condition. This expression pattern of the *EPAS1* gene in Tibetan sheep is similar to that in humans; for example, under plateau hypoxic conditions, the plain population can adapt to hypoxia rapidly by increasing the expression of *EPAS1* and altering the conformation of hemoglobin [34]. 

This study found that among the six tissues of Tibetan sheep at three different altitude gradients, the expression of *EPAS1* was the highest in the heart. The heart is an aerobic organ that relies on a continuous supply of oxygen to produce high-energy phosphate ATP for mechanical functions, the high expression of *EPAS1* in this organ confirm that this gene related with oxygen metabolism. The increased expression of *EPAS1* in individual tissues of plateau animals suggested that it forms the molecular basis for maintaining the normal functioning of biological activities in a low-oxygen environment. These results suggest that *EPAS1* can enhance the adaptability of Tibetan sheep to a low-oxygen environment by increasing its expression. In addition, the experimental results provide a theoretical basis for further elucidating the mechanism of hypoxia adaptation in mammals.

## 5. Conclusions

Six SNPs were detected in four *EPAS1* alleles, one of which was non-identical polymorphism (P.f606L); the distribution of these four alleles was significantly different between Tibetan sheep and Hu sheep. One region of variation was associated with increased *p*O_2_ and sO_2_, suggesting that the variation in the *EPAS1* gene is related to oxygen metabolism in sheep. The expression of the *EPAS1* gene was positively correlated with the altitude in six tissues; the expression of *EPAS1* was the highest in the heart and the lowest in the spleen. In summary, these results suggested that the expression and mutation in the *EPAS1* gene are associated with oxygen metabolism in sheep.

## Figures and Tables

**Figure 1 genes-13-01871-f001:**
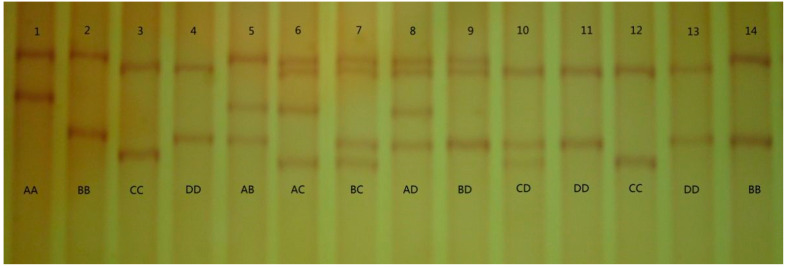
PCR-SSCP analysis of sheep *EPAS1*. Different banding patterns came about from paring of four unique variant sequences (*A* to *D*) in either homozygous or heterozygous forms.

**Figure 2 genes-13-01871-f002:**
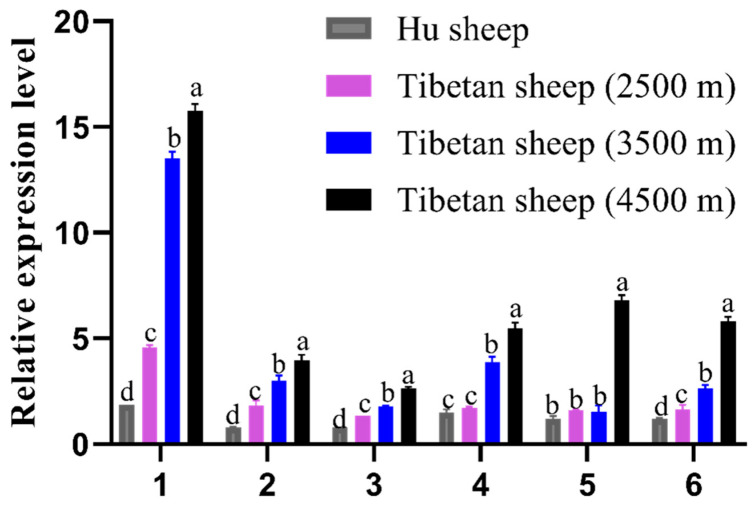
Relative expression levels of *EPAS1* mRNA in different tissues of Hu sheep and Tibetan sheep from different altitudes. Heart (1), liver (2), spleen (3), lung (4), kidney (5), and longissimus dorsi muscle (6) tissue were analyzed for mRNA expression. The bars represent the mean ± SE from three independent biological replicates, each performed with three technical replicates. The different letters indicate significant differences (*p* < 0.05); the same letter indicates no significant differences (*p* > 0.05).

**Table 1 genes-13-01871-t001:** Primers for PCR and RT-qPCR.

Gene	Primer Sequence (5′ → 3′)	Product Size (bp)	Annealing Temperature/°C	Purpose
*EPAS1*	F: GAGAGGACTTCCAGCTTAGC R: TCCCCTATGCAGCCCAGC	470	61	PCR
*EPAS1*	F: CCTCTCACAGCACCTTCCTC R: GCCACTGTGTGCTGGATATG	122	60	RT-qPCR
*β-actin*	F: AGCCTTCCTTCCTGGGCATGGA R: GGACAGCACCGTGTTGGCGTAGA	113	60	RT-qPCR

**Table 2 genes-13-01871-t002:** Sequence variation identified in ovine *EPAS1*.

Location	*A*	*B*	*C*	*D*	Ammo Acid Change
c.1641	G	G	A	A	no change
c.1674	G	G	C	G	no change
c.1816	T	T	C	T	p.F606L
c.1824	T	C	C	C	no change
c.2028	G	G	A	G	no change
c.2039+14	G	G	A	G	no change

**Table 3 genes-13-01871-t003:** Genotype frequencies of *EPAS1* exon15 in sheep.

Breeds	*n*	Genotype Frequency (%)									
		*AA*	*BB*	*CC*	*DD*	*AB*	*AC*	*AD*	*BC*	*BD*	*CD*
Tibetan sheep	332	0.30	83.13	0	0	7.23	0.9	0.3	3.31	4.22	0.60
Hu sheep	339	8.85	9.73	1.77	5.01	16.22	7.96	13.57	12.98	15.93	7.96

**Table 4 genes-13-01871-t004:** The effect of breed and gender on blood gas.

		*p*CO_2_	*p*O_2_	sO_2_	*p* _50_
breeds	Tibetan sheep (*n* = 176)	39.819 ± 0.926 ^B^	34.820 ± 1.050 ^B^	64.710 ± 1.430 ^B^	26.435 ± 0.129
	Hu sheep (*n* = 231)	42.235 ± 0.592 ^A^	42.212 ± 0.669 ^A^	76.553 ± 0.916 ^A^	26.615 ± 0.082
^1^ Gender	Male (*n* = 134)	39.996 ± 0.880 ^B^	37.804 ± 0.994	70.400 ± 1.360	26.532 ± 0.122
	Female (*n* = 42)	42.058 ± 0.589 ^A^	39.227 ± 0.667	70.862 ± 0.913	26.519 ± 0.081

^1^ Gender refers to Tibetan ewes and Tibetan rams. Breed and gender were used as fixed factors in the model to evaluate the effect of gene variation on *p*CO_2_, and the breed was used as the fixed factor in the model for other association analyses. Not sharing an uppercase superscript (A or B) indicates significant differences (*p* < 0.01).

**Table 5 genes-13-01871-t005:** Association of *EPAS1* variants with blood gas (mean ± SE) ^1^ in sheep.

Blood Gas	Variant	Absent		Present		*p* ^1^
		Mean ± SE	*n*	Mean ± SE	*n*	
*p*CO_2_	*A*	40.416 ± 0.454	271	40.830 ± 0.645	136	0.568
	*B*	41.530 ± 0.730	107	40.253 ± 0.435	300	0.107
	*C*	40.497 ± 0.417	328	40.783 ± 0.798	79	0.726
	*D*	40.528 ± 0.436	306	40.583 ± 0.708	101	0.943
*p*O_2_	** *A* **	**37.517 ± 0.509**	**271**	**39.514 ± 0.725**	**135**	**0.015**
	*B*	38.050 ± 0.833	106	38.133 ± 0.493	300	0.927
	*C*	38.250 ± 0.471	327	37.364 ± 0.900	79	0.337
	*D*	38.417 ± 0.491	305	37.114 ± 0.799	101	0.132
sO_2_	** *A* **	**64.639 ± 0.907**	**271**	**71.798 ± 0.991**	**135**	**0.039**
	*B*	70.160 ± 1.140	106	70.187 ± 0.672	300	0.983
	*C*	70.251 ± 0.643	327	69.800 ± 1.230	79	0.717
	*D*	70.449 ± 0.671	305	69.300 ± 1.090	101	0.329
*p* _50_	*A*	26.695 ± 0.063	271	26.537 ± 0.089	135	0.133
	*B*	26.564 ± 0.102	106	26.671 ± 0.060	300	0.329
	*C*	26.640 ± 0.058	327	26.689 ± 0.110	79	0.666
	*D*	26.662 ± 0.060	305	26.601 ± 0.098	101	0.567

^1^ Estimated marginal means and standard errors (SE) of those means derived from general linear mixed-effects models that included “breeds” and “gender” as fixed factors. *p* < 0.05 are in bold.

## Data Availability

The authors affirm that all data necessary for confirming the conclusions of the article are present within the article, figures, and tables.

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
