# Peer review of "Expression and Variations in EPAS1 Associated with Oxygen Metabolism in Sheep"

_genes, 2022, doi:10.3390/genes13101871_

Round 1

Reviewer 1 Report

The current manuscript has a high originality. The experimental design is fitting with the objectives and the research methodology utilized relevant and properly administered. In addition, methods of statistical data analysis is adequate. The results is well presented and well discussed with recent and updated related literature. The overall evaluation of the current manuscript is to accept it for publication with minor English language revision. Indeed, the manuscript focuses on an interesting topic of in sheep genomics that is related to journal scope.

Reviewer 2 Report

The study was carried out with a sufficient number of animals using the appropriate method. The results are written clearly in a way that the reader can understand. Therefore, the presented article can be published in its current form.
